# Constant strain accumulation rate between major earthquakes on the North Anatolian Fault

Ekbal Hussain [1], Tim J. Wright[1], Richard J. Walters[2], David P.S. Bekaert[3], Ryan Lloyd[4] & Andrew Hooper[1]

Earthquakes are caused by the release of tectonic strain accumulated between events. Recent advances in satellite geodesy mean we can now measure this interseismic strain accumulation with a high degree of accuracy. But it remains unclear how to interpret short-term geodetic observations, measured over decades, when estimating the seismic hazard of faults accumulating strain over centuries. Here, we show that strain accumulation rates calculated from geodetic measurements around a major transform fault are constant for its entire 250-year interseismic period, except in the ~10 years following an earthquake. The shear strain rate history requires a weak fault zone embedded within a strong lower crust with viscosity greater than ~$10^{20}$ Pa s. The results support the notion that short-term geodetic observations can directly contribute to long-term seismic hazard assessment and suggest that lower-crustal viscosities derived from postseismic studies are not representative of the lower crust at all spatial and temporal scales.

[1] COMET, School of Earth and Environment, University of Leeds, Leeds LS2 9JT, UK. [2] COMET, Department of Earth Sciences, Durham University, Durham DH1 3LE, UK. [3] Jet Propulsion Laboratory, California Institute of Technology, Pasadena, CA 91125, USA. [4] COMET, School of Earth Sciences, University of Bristol, Bristol BS8 1TH, UK. Correspondence and requests for materials should be addressed to E.H. (email: e.hussain@leeds.ac.uk)

One of the primary inputs into any probabilistic seismic hazard assessment (PSHA) model is a catalogue of earthquake sources that have occurred in the past[1, 2]. A major problem with this approach is that catalogues are usually incomplete, as the time between earthquakes often greatly exceeds the catalogue length[3]. With recent improvements in the accuracy and spatial coverage of geodetic observations from GNSS and InSAR[4, 5], it has been proposed that measurements of inter-seismic strain may complement or even supersede traditional PSHA methods[6, 7]. This approach is supported by the broad agreement between geodetic strain rates and seismicity rates in California and Turkey[4].

However, in 2-layer linear Maxwell viscoelastic crustal models of the earthquake deformation cycle, which are commonly used to interpret interseismic deformation, strain rate varies as a function of time between earthquakes[8-10], with the shear strain rate in the fault zone decreasing with time since the earthquake. If this is true then short-term geodetic estimates of surface strain accumulation rate may give a biased estimate of the long-term strain rate; observations close in time after an earthquake will overestimate the long-term strain rate, and hence the seismic hazard and observations a long time after an earthquake will underestimate the long-term strain rate[11, 12]. Alternative models of the earthquake cycle incoporate rate-and-state friction methodologies developed from labaoratory experiments to explain geodetic observations from specific parts of the earthquake cycle[13-15], while others employ more complex crustal viscoelastic rheologies to explain the early postseismic and late interseismic observations, including non-Newtonian power law models[16] or Burger's body rheologies[17]. The evolution of strain rate through the entire inter-event period provides a powerful test of such models.

The long inter-event time in many fault zones, typically hundreds to thousands of years, means we do not have deformation observations spanning a complete inter-event period for most faults[11]. Here, we build an inter-event strain history for a single fault zone by using measurements of strain rate from different portions of the North Anatolian Fault (NAF) in Turkey, where the most recent earthquake for each portion has occurred at different times[18, 19]. In the last 80 years the NAF has failed in 10 large earthquakes (Mw > 6.5), which have ruptured over 1000 km of the fault with an average slip of ~2–5 m[19]. If we assume rheological properties are similar along the entire fault, we can build a strain rate history sampling the majority of the ~250-year inter-event period on the NAF by using geodetic measurements of strain rate in the location of each of these previous ruptures, along with GNSS observations collected before the 1999 Izmit earthquake, ~245 years after the previous major earthquake in 1754[19]. Our results show that strain accumulation reaches near steady state within ~10 years of an earthquake. We discuss the implications for seismic hazard assessment and the rheology of continental lithosphere.

## Results

**Geodetic measurements of strain accumulation**. We mapped the surface deformation along the entire subaerial expression of the NAF (~1000 km) with InSAR using satellite radar data from the European Space Agency's Envisat mission. Our dataset consists of a total of 608 Synthetic Aperture Radar (SAR) images from 14 descending and 9 ascending satellite tracks that span the time interval between 2002 and 2010 (Fig. 1a, b, Supplementary Figs. 1, 2). We processed the data to obtain average satellite line-of-sight (LOS) velocities using methods described in Hussain et al.[20]. Descending data are complete for the entire fault. Ascending data are complete except for a gap between about

35° E and 37° E (Fig. 1b), where insufficient acquisitions were made for us to obtain reliable velocities. Further details of the data processing for each track are given in Supplementary Table 1 and the Methods section.

To estimate the uncertainties in the LOS data we calculate the RMS misfit in velocities in the overlapping areas between neighbouring tracks, after projection into horizontal velocities using the local incidence angles[20] (Supplementary Fig. 3). The residuals between neighbouring tracks are approximately Gaussian with mean values close to zero. The average RMS misfits between these independent estimates of horizontal velocities are 4.4 mm yr$^{-1}$ for descending tracks and 5.4 mm yr$^{-1}$ for ascending tracks, giving empirical uncertainties of ~3 and ~4 mm yr$^{-1}$, respectively, for the individual tracks in the horizontal and an uncertainty of 1.2–1.6 mm yr$^{-1}$ in the LOS.

We transform the estimated LOS velocities for each track from a local reference north of the fault (an average of pixels in a 2 km radius), into a Eurasia-fixed reference frame by first resampling the InSAR LOS velocities onto a 1 km by 1 km regular grid. For each track, we then determine the best-fit plane between the GNSS velocities projected into the LOS and the InSAR velocities within 1 km of each GNSS site, and remove this from the InSAR velocity maps. For pixels with both ascending and descending LOS velocities, we invert for the east-west and vertical components of motion using the smooth, interpolated north component of the GNSS velocities (Supplementary Fig. 4) to constrain the north-south component in the inversion[20]. Using a smooth north-south velocity field does not lead to smoothed east-west velocities in the inversion because the LOS is not very sensitive to the north component. For the areas with LOS data from only a single geometry we also assume no vertical motion.

**An interseismic strain history for the NAF**. Our resulting east-west velocity field (Fig. 1c) clearly shows a north-south gradient in east-west velocity across the NAF, consistent with strain accumulation along the entire NAF with the expected right-lateral sense of motion. There is no systematic pattern in vertical velocities across the fault (Supplementary Fig. 5).

To investigate the spatial variation in strain accumulation along the fault we plot profiles of fault parallel velocity at regular intervals (every 1/2 degree). The profiles show a remarkably consistent pattern along the entire fault, with the transition from Eurasian velocities in the North to Anatolian Velocities in the south occurring over a region that is ~70 km wide. The exception is in the two regions where fault creep is known to occur, at Ismetpasa and Izmit[14, 21, 22]. Here, there is a sharp step in east-west velocity across the fault superimposed upon the broader strain accumulation signal, consistent with previous interpretations that the shallow part of the fault is creeping at a rate less than the plate loading rate.

There is considerable local scatter in the east-west velocity field which prevents us from estimating the strain rate directly from the data. Instead we use a simple arctangent functional fit through the InSAR and GNSS velocities, based on the analytical solution to an infinitely long screw dislocation in an elastic half space[23]. This function has two parameters: the slip rate, which is an estimate of the far-field change in velocity between Anatolia and Eurasia, and the locking depth, which is dependent on the length-scale of the transition. The surface strain rate at the fault is proportional to the slip rate and inversely proportional to the locking depth (see Methods section for details). Note that we also account for the rotation of Anatolia and, in the areas of creep, we solve for the shallow creep rate and depth using a simple elastic dislocation model[20], so that we can remove this contribution to the strain. The simple dislocation model (see Methods) is

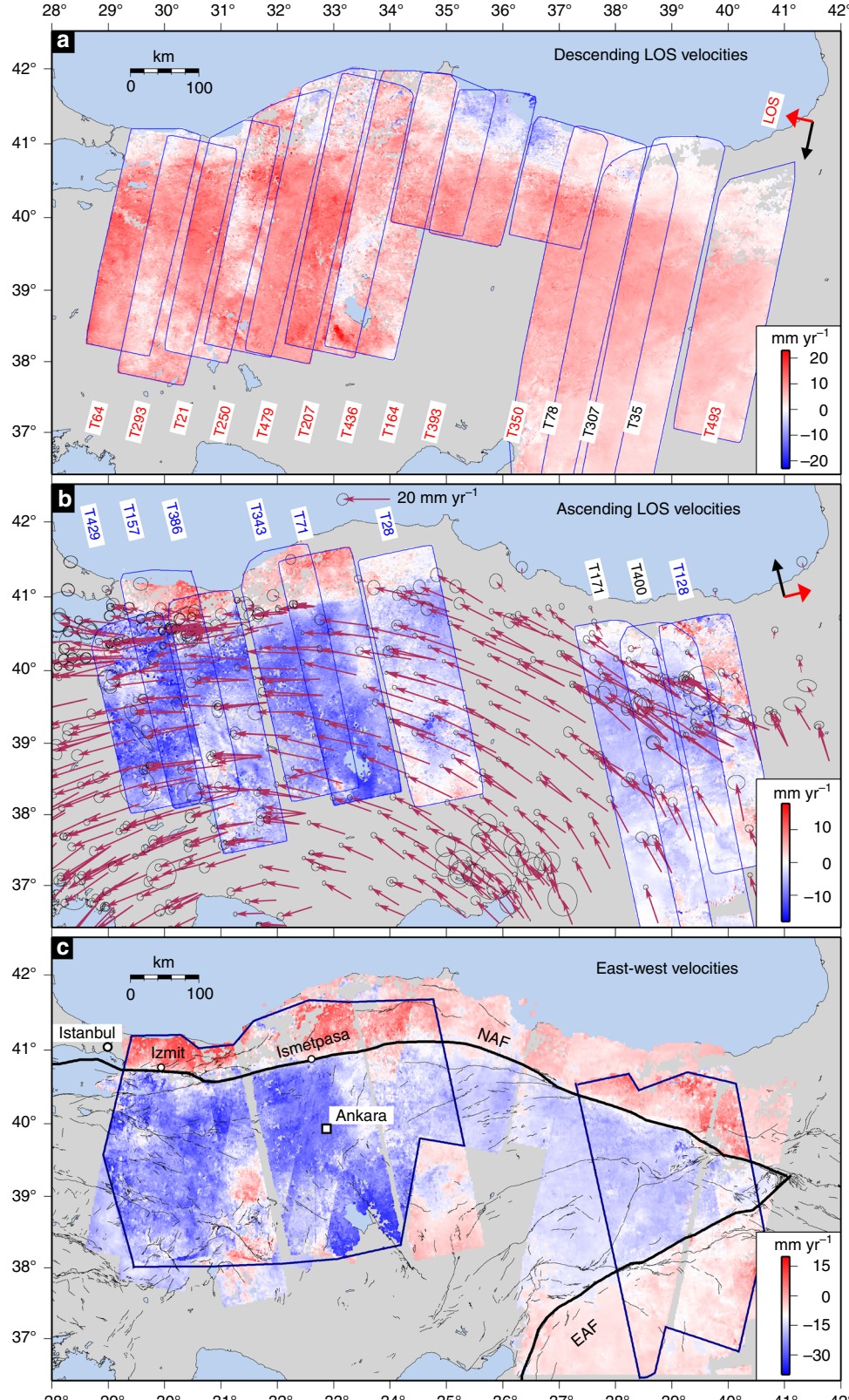

**Fig. 1** InSAR derived horizontal velocity field. Satellite line-of-sight (LOS) velocities for northern and eastern Turkey. LOS velocities are relative to Eurasia for the descending (**a**) and ascending (**b**) tracks used in this study. Tracks labelled in black in **a** and **b** were processed by Walters et al.[58]. Red colours show motion away from the satellite. The maroon vectors are published GNSS velocities from the Global Strain Rate Model[61]. **c** The east-west component of motion, relative to Eurasia, decomposed from the LOS measurements and the interpolated GNSS north velocities; see text for details. White in the colour scale is set at −10 mm yr$^{-1}$ to emphasise the change in velocity across the fault. Negative velocities show motion towards the west. The bold black lines indicate the main strands of the North Anatolian Fault (NAF) and the East Anatolian Fault (EAF). The polygons indicate regions with both ascending and descending data. The pale regions outside the polygons are covered by only ascending or descending data

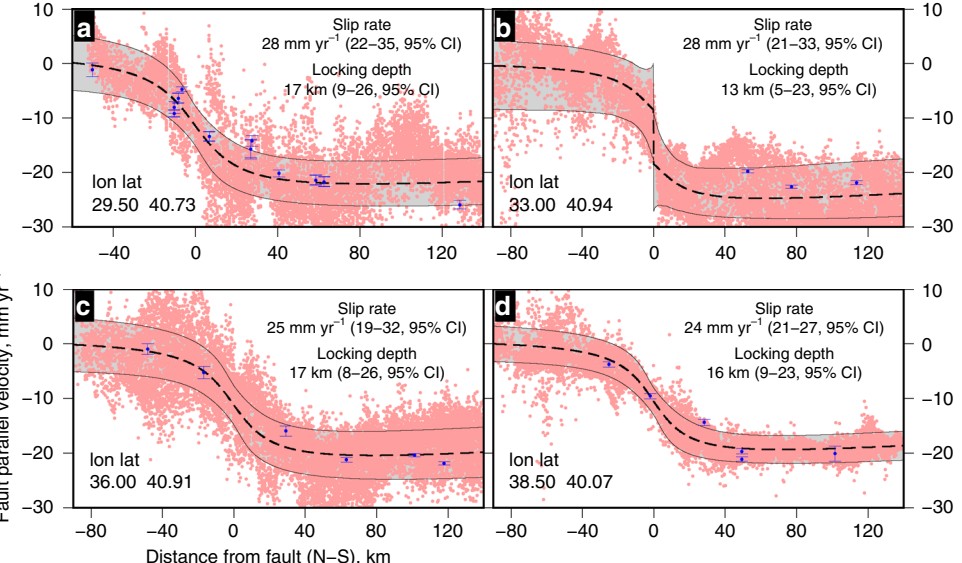

**Fig. 2** Estimating fault parameters from velocity profiles. A selection of profiles (**a–d**) used to estimate the fault slip rate and locking depth (red points in Fig. 3) along the North Anatolian Fault. All 22 profiles are shown in Supplementary Fig. 6. The red points are the fault parallel component of the horizontal velocity field projected from within 25 km perpendicular distance onto each profile. The blue points are the fault parallel component of the GNSS velocities. The black dashed line is our maximum a posteriori probability (MAP) solution with the grey shaded area representing the 95% confidence interval

sufficient to account for the effect of aseismic creep in this case because the creep is generally limited to shallow portions of the locked fault (≤~5 km) causing deformation with a spatial wavelength of about 10 km[22]. In comparison, the deep interseismic strain, the main focus of this study, has a much broader deformation signal, ~100 km wide. Additionally, the profiles through the velocities (Supplementary Fig. 6) show that any deviation from zero north of the fault is likely the result of atmospheric or other noise in the InSAR data.

The results (Figs. 2, 3 and Supplementary Figs. 6, 7) show the variation in slip rate, locking depth and hence surface strain rate along the NAF. We see a general pattern of westward increasing slip rates from an average ~22 ± 3 mm yr$^{-1}$ on the eastern section of the NAF to ~30 ± 3 mm yr$^{-1}$ in the west (Fig. 3b). This increase is due to internal deformation (east-west extension) in Anatolia[24] and can clearly be seen by comparing fault-perpendicular profiles of GPS velocities (Supplementary Fig. 7). We correct for this along-strike variation in slip-rate (see Methods) to ensure that any residual variation in strain-rate can be compared directly to the time since the last earthquake.

In general, our maximum a posteriori probability (MAP) solutions for the locking depth show no clear systematic variation along strike. The two higher estimates of the locking depth along the Izmit portion could represent a variation in the locking depth through the earthquake cycle[25], but this is complicated by the fact that the NAF in this region breaks into multiple strands, thus widening the zone of strain accumulation[26]. The high locking depth in both pre-1999 (Supplementary Fig. 8) and post-1999 profiles support the fact that the strain has split onto multiple strands. Although slip rate and locking depth estimates co-vary, the slip rates are not greatly affected by locking depth in this case (Supplementary Fig. 9).

If we assume no internal deformation within central Turkey then the projection of far field GNSS velocities onto the fault also gives the estimated slip rate from GNSS alone with no required prior assumption on the deformation model. These velocities are indicated by the purple lines in Fig. 3b for five broad profiles (~150 km wide, Supplementary Fig. 10), which show good agreement with the slip rates derived from the velocity field.

The fact that slip rate increases from east to west necessitates internal deformation of Anatolia, consistent with the estimates from GNSS[24] for east-west extension within Anatolia.

The estimated surface strain rates at the fault (Fig. 3d) are remarkably constant along the fault, with a value of ~0.5 microstrain yr$^{-1}$. There is no clear spatial correlation in slip rate, locking depth, or strain rate with the location of previous large ruptures along the NAF.

If we assume that the rheological properties are similar along the fault, we can plot the strain rate as a function of time since the most recent earthquake (Fig. 4). Our results derived from Envisat cover the period from 10 years to 85 years following an earthquake. We add an additional measurement at 245 years post-earthquake by assessing the slip rate, locking depth, and strain rate from GNSS data acquired before the 1999 earthquakes where the previous earthquake had occurred in 1754[27] (Supplementary Fig. 8). We also estimate surface strain rates from GNSS observations for the first 7 years of the postseismic period using data collected following the 1999 earthquakes[28] (see Methods). Collectively, this strain rate history spans the majority of the ~250-year period in Turkey. We make a small correction to the strain rates for the internal extension of Anatolia using the far-field GNSS measurements, normalising to an average slip rate of 26 mm yr$^{-1}$ (see Methods).

**Implications for seismic hazard assessment.** The results (Fig. 4) show that interseismic strain rate is independent of time since the most recent earthquake, once a ~10-year postseismic transient period has passed. If this result holds for other fault zones, short-term observations of present-day tectonic strain accumulation are representative of long-term deformation rates. Geodetic strain rates could therefore be used as a measure of future seismic hazard[7].

It is clear that significant displacement occurs during the postseismic transient, and therefore an assessment of the total strain accumulation would require knowledge (or a model) of the deformation pattern during the early stages of the interseismic period. This depends on the magnitude of the postseismic strain transient and the inter-event time interval. For example, the total

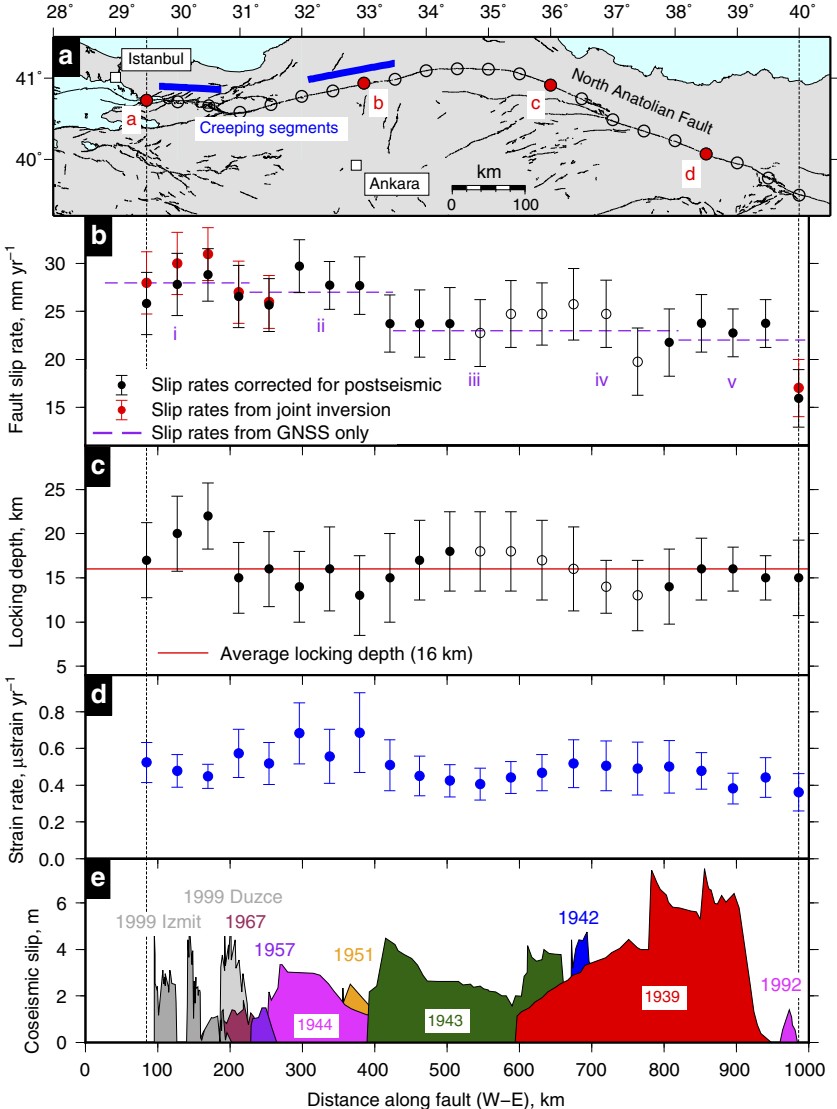

**Fig. 3** Along strike variation in fault parameters. The variation in fault slip rate (**b**) and locking depth (**c**) along strike of the North Anatolian Fault (NAF) at the locations indicated by the black circles in **a**. The error bars represent the 68% confidence bound on the parameter estimates. The solid circles are results from profiles that are in the high confidence polygons indicated in Fig. 1c, while the open circles are profiles that are in areas where only ascending or descending data are available. The slip rates are consistent with a constant locking depth of 16 ± 4 km along the entire fault (Supplementary Fig. 9). The purple lines in **b** are the slip rate estimates from GNSS alone (Supplementary Fig. 10). We use the slip rate and locking depth estimates to calculate the strain rate along the fault (**d**), as described in the text. **e** The surface coseismic slip distributions of major earthquakes (Mw > 6.5) along the NAF since 1939[19, 62, 63]

strain due to the ~10-year postseismic transient in Fig. 4 is about 10% of the total strain accumulation in the 250 interseismic period due to long-term loading. For long inter-event periods the fraction of the total strain due to postseismic deformation would be smaller, and thus the impact on the hazard estimate would be minimal. However, the opposite is true for short inter-event periods. While a $1/t$ decay of postseismic velocities appears to be a common feature of many earthquakes globally[29], there are significant variations in the magnitude of postseismic signals and hence the duration for which they make a significant impact. In a global compilation of postseismic deformation[29], the fastest postseismic transients were found to occur at a rate that is ~2.5 times larger than those observed after the 1999 earthquakes on the NAF. This rapid rate of postseismic deformation would account for about 21% of the total strain accumulation for a 250-year period.

Translating interseismic strain rates into forecasts of seismic hazard, for example, via a PSHA assessment, is not straightforward,

but several approaches have recently been proposed[6, 30–32]. A major assumption necessary for translating geodetic strain to seismic strain is the proportion of strain that is released aseismically by fault creep, slow slip, or plastic deformation[4]. Other required assumptions include the Gutenberg-Richter $b$-value, the expected maximum magnitude, the seismogenic thickness and the type of the earthquake mechanism that might occur. At present earthquake forecasts from geodesy require calibration against historical seismicity to determine how the proportion of aseismic strain release varies across different tectonic regions[7]. Since the long-term interseismic deformation occurs at a long wavelength relative to the short wavelength shallow creep signal[22], it might be possible to disentangle the two signals along creeping sections[33], or use auxiliary strain measurements from creep metres or strain metres to account for the aseismic creep signal[34].

In regions where very large earthquakes occur with low frequency (inter-event intervals of centuries to millenia), PSHA

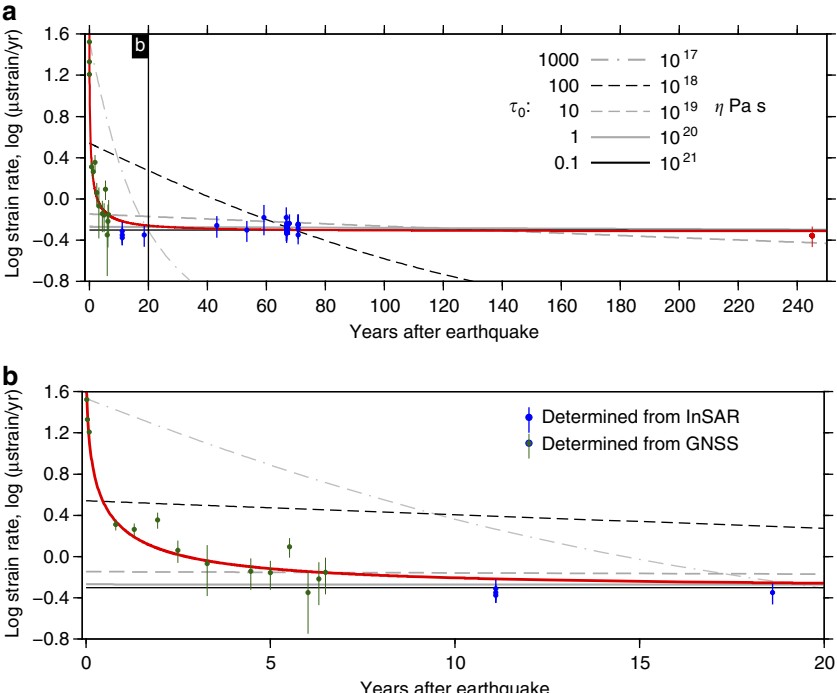

**Fig. 4** An interseismic strain rate history. **a** The shear strain rates calculated using our estimates of the locking depths and slip rates (Fig. 3d), and corrected for the internal extension of Analtolia, plotted against the time since the last earthquake, in blue. The red circle is the estimate using pre-1999 earthquake GNSS measurements[27]. The dark green points are the postseismic strain rates estimated from the displacement time series time series of two GNSS stations located ~10 km either side of the fault (SEFI and KAZI)[28]. The red line is the best fit to the postseismic and interseismic GNSS strain rates (see text). The black and grey lines are the expected temporal variation in strain rate for different lower crustal viscosities away from the fault plane[8, 10]. $\tau_0$ values of 1000, 100, 10, 1, and 0.1 correspond to lower crustal viscosities of ~$10^{17}$, ~$10^{18}$, ~$10^{19}$, ~$10^{20}$, and ~$10^{21}$ Pa s, respectively. **b** A blow up of the same figure for the postseismic phase. Error bounds represent 1 sigma

models are reliant on catalogues of historical seismicity, which are often incomplete. This is demonstrated in the Himalayas, where geodetic results predict that larger earthquakes are required to close the strain budget[35], but these missing earthquakes are not yet accounted for in current PSHA models.

**Rheology of the lower crust and upper mantle**. Our inter-event strain rate history for the NAF also places strong constraints on the rheology of the lower crust and mantle. A simple yet instructive model that attempts to predict strain rates as a function of time for a strike-slip fault is the viscoelastic-coupling model, in which repeating earthquakes occur in an elastic layer overlying a uniform linear-Maxwell viscoelastic half space[8, 9, 36]. While this model does not represent the full complexity and spatial variability of potential rheologies, its relative simplicity allows us to understand the broad constraints that the observations place on the properties of the lithosphere in and around fault zones.

The key parameter that controls the temporal behaviour of these models is $\tau_0$, the ratio of inter-event time ($T$) to the Maxwell relaxation time of the viscoelastic substrate ($2\eta/\mu$), where $\eta$ is the viscosity and $\mu$ is the shear modulus. Models with $\tau_0 \gg 1$, i.e., Maxwell relaxation time $\ll$ inter-event time, predict a rapidly decreasing strain rate with time, while models with $\tau_0 \lesssim 1$ predict a nearly constant strain rate between earthquakes (Fig. 4).

We compare our observations at the NAF with predictions of the strain rate variation with time for different substrate viscosities using $T = 250$ years[19] and $\mu = 3 \times 10^{10}$ Pa (Fig. 4). We fix the thickness of the elastic lid, which ruptures completely in each earthquake, to 16 km. In the long term the upper crustal blocks slide past each other at a geodetically determined rate of 26 mm yr$^{-1}$[26, 37], which is at the upper end of geological

estimates of the slip rate[38–40]. We calculate strain rate histories for $\tau_0$ values of 1000, 100, 10, 1, and 0.1, corresponding to average viscosities in the viscoelastic substrate (lower crust and mantle) of ~$10^{17}$, ~$10^{18}$, ~$10^{19}$, ~$10^{20}$, and ~$10^{21}$ Pa s, respectively, using the equations given in Appendix A of Savage[10].

There are two key observations from the NAF that models of the earthquake deformation cycle must match—the long-term invariance of strain rate and the rapid decay of postseismic strain. Each provides important constraints on different parts of the system; we argue below that the rheology of the substrate away from the fault controls the long-term interseismic strain, and the rheology of the fault zone itself controls the temporal decay of postseismic deformation.

Viscoelastic coupling models with high $\tau_0$ cannot explain time-invariance of strain rate that we observe following the initial postseismic period. To obtain the observed strain rates at long times after an earthquake requires relaxation times that are approximately equal to or longer than the inter-event time—long-lived focused interseismic strain in the viscoelastic coupling model is really postseismic deformation that has yet to decay. Therefore, for the NAF, this translates to a high long-term viscosity of the substrate of $\gtrsim 10^{20}$ Pa s.

The requirement for viscosities away from the fault to be high in order to match interseismic strain observations is consistent with more complex viscous models that explicitly separate out the viscosity of the fault zone from that of the substrate[12], or models in which the effective viscosity of the fault zone is reduced through shear-heating and non-Newtonian effects[16]. Models in which the entire earthquake cycle is explained through frictional processes embedded within an elastic crust[41] are of course also consistent with a high viscosity away from the fault. A lower crust that, away from fault zones, relaxes on a timescale that is long

compared to the inter-event time is an inescapable inference of the widespread observation of focused interseismic strain[11, 42], and an essential requirement when focused interseismic strain is present late in the earthquake cycle[4]. Simply put, if the lower crust relaxes on a short timescale, there is no plausible reason for interseismic strain to focus around the upper-crustal fault.

For the postseismic period, no single value of $\tau_0$ can explain the observed evolution of strain that dominates the geodetic signals for the first few years following a large earthquake. The viscoelastic coupling model, with a uniform linear Maxwell rheology in the substrate, predicts an exponential decay in postseismic velocities (and strain rate). The effective viscosity required to match the observations would need to increase with time. Ingleby and Wright[29] compiled geodetic observations of postseismic velocities from all continental earthquakes worldwide and showed that they decayed as a function of $1/t$. A $1/t$ decay in strain rate fits the observations of the NAF (red line in Fig. 4).

A $1/t$ decay in postseismic strain rate is consistent with postseismic strain being driven entirely by rate and state frictional afterslip[29], by viscoelastic relaxation of a non-Newtonian power-law material with a high stress exponent[43], or by relaxation of a substrate with multiple relaxation times, for example where viscosity decreases as a function of depth[44, 45]. Models that explicitly test the size of the region that relaxes in the postseismic period show that it must be confined both laterally and in its depth extent[12, 46]. Moore and Parsons[47] showed that the localisation of shear in a narrow zone beneath a strike slip fault is a natural consequence of realistic substrate properties—depth-dependent viscosity, shear heating, and non-Newtonian effects. Postseismic deformation therefore results from the relaxation of material in the fault zone itself, embedded in a stronger substrate.

Previous studies have used the depth distribution of earthquakes in the continents to suggest that the strength of the lithosphere resides in the upper crust[48]. However, our analysis of the strain rate shows that the lower crust away from the fault zone must also be relatively strong. Furthermore, the evolution of postseismic deformation suggests that the presence of fault zones modifies the local rheology[16] rather than requiring the entire lower crust to have a low viscosity. Seismogenic thickness may therefore not be a useful proxy for crustal strength at major fault zones.

## Methods

**InSAR data processing**. We process the InSAR data following the methods described in Hussain et al.[20]. We focus the Envisat SAR images using ROIPAC[49] and use the DORIS software[50] to construct interferograms that minimise the temporal and perpendicular baselines while producing a redundant network for each track (Supplementary Figs. 1, 2). We correct for topographic contributions to the radar phase using the 90 m SRTM Digital Elevation Model[51] and account for the oscillator drift for Envisat[52]. We unwrap the interferograms using an iterative unwrapping procedure for small baseline InSAR measurements described in Hussain et al.[20]. We correct each interferogram for an estimate of the tropospheric noise using auxiliary data from the ERA-Interim global atmospheric model reanalysis product[53, 54]. On average the ERA-I correction reduces the standard deviation of phase within our tracks by about 5% (Supplementary Table 1). We use the StaMPS (Stanford Method for Persistent Scatterers) small baseline time series technique[55, 56] to remove incoherent pixels and reduce the noise contribution to the deformation signal, and to calculate the average LOS velocity for each track. We present 1-sigma uncertainties on the final velocities for each pixel, estimated using bootstrap resampling[57].

Our InSAR dataset includes five tracks published by Walters et al.[58] (descending tracks 78, 307 and 35, and ascending tracks 171 and 400), and an additional track that was previously unpublished (descending track 493), which cover the eastern section of the NAF (Fig. 1). The interferograms for these tracks were created using ROIPAC, with the InSAR corrections applied as discussed above, and the velocity maps formed using the $\pi$-RATE software package[59]. The main difference between $\pi$-RATE and StaMPS is related to the selection of the pixels, while the mathematical expression for the rate-computation does not change. See the original paper[58] for more details on the processing of these tracks.

**Modelling profiles**. We fit a simple 1-D elastic dislocation model[20] to the fault parallel velocities ($v_{par}$), using a screw dislocation model (Eq. (1)) for most of the fault to solve for slip rate ($S$) and locking depth ($d_1$). For creeping sections (see

Fig. 3) we also solve for the creep rate ($C$) and creep depth ($d_2$) (Eq. (2)).

$$v_{par}(x) = \frac{S}{\pi}\arctan\left(\frac{x}{d_1}\right) + x\theta_{rot} + a, \qquad (1)$$

$$v_{par}(x) = \frac{S}{\pi}\arctan\left(\frac{x}{d_1}\right) + C\left[\frac{1}{\pi}\arctan\left(\frac{x}{d_2}\right) - \mathcal{H}(x)\right] + x\theta_{rot} + a, \qquad (2)$$

where $a$ is a static offset, $x$ is the perpendicular distance to the fault, $\mathcal{H}(x)$ is the Heaviside function, and $\theta_{rot}$ corrects for the proximity of the profile points to the pole of rotation of Anatolia in a Eurasia-fixed reference frame. $\theta_{rot}$ is calculated using the linear trend through the far-field GNSS velocities on five broad profiles (Supplementary Fig. 10), and assuming the pole of rotation is fixed at the location found by Reilinger et al.[26]. The values used and the longitude extent to which they apply are given in Supplementary Table 2.

We find the best-fit values for each model parameter using a Markov Chain Monte Carlo (MCMC) Bayesian sampler[22, 60]. The MCMC sampler explores the parameter space constrained by: $-60 < S$ (mm yr$^{-1}$) $< 0$, $0 < d_1$ (km), $<60$, $-30 < C$ (km), $<0$, $0 < d_2$ (km), $<40$, $-40 < a$ (mm yr$^{-1}$) $< 40$, assuming a uniform prior probability distribution over each range. For creeping profiles an important constraint we impose is that the maximum creep depth cannot be greater than the locking depth, i.e., $d_2 \leq d_1$. Our MCMC model runs over 300,000 iterations and produces 48,000 samples of the posterior distribution from which we estimate both the MAP solution and marginalised probability distributions for each parameter.

**Calculating strain rates**. Differentiating Eq. (1) and setting $x = 0$ gives the surface shear strain rate at the fault:

$$\dot{\epsilon} = \frac{S}{2d_1}. \qquad (3)$$

We use Eq. (3) to calculate the strain rate at the fault for each of our profiles ensuring we propagate the full covariance information for the slip rate and locking depths.

In Fig. 3b we showed that the slip rates increase from an average ~22 ± 3 mm yr$^{-1}$ on the eastern section of the NAF to ~30 ± 3 mm yr$^{-1}$ in the west. Most of this increase is related to the east-west extension within central and western Anatolia[24]. This is an overall feature of the large-scale deformation field in Turkey and is not related to time since last earthquake. Therefore, we need to correct for this effect before comparing the strain rates from different positions along the fault.

We do this by using assuming that the far-field GNSS velocities inform us about the large-scale spatial changes in slip-rate independent of inter-seismic deformation on the fault, whereas the InSAR velocites inform us about these spatial changes and also temporal changes associated with inter-seismic deformation. Therefore, the far-field GNSS slip rates estimated in Supplementary Fig. 10, can be used to correct for the large-scale deformation signal.

We calculate the difference between the GNSS slip rate and an average slip rate of 26 mm yr$^{-1}$, i.e., $\Delta s = s_i - s_0$, where $s_0 = 26$ mm yr$^{-1}$ and $s_i$ is the slip rate estimated from our inversion. When calculating the strain rate for plotting in the comparison Fig. 4, we use ($s_i - \Delta s$) instead of $s_i$. This results in an average change of 5%, which has minimal effect on our interpretation.

We calculated the postseismic strain rates after the 1999 Izmit earthquake using the GNSS time series recorded for 7 years following the earthquake[28]. To do this, we calculated the relative displacement time series between two stations located 15–20 km either side of the fault (KAZI and SEFI) and divided by the distance between the two stations. We are confident that the postseismic strain signal recorded between these stations reflect the deeper depth-average afterslip rather than the shallow creep on the fault, because previous work[22] has shown that the spatial wavelength of deformation due to aseismic creep on the Izmit portion of the NAF is mostly constrained to around 5–10 km either side of the fault.

**Data availability**. The SAR data from the Envisat satellite mission, used in this study, are available to download for free from the European Space Agency's Virtual Archive 4 website: http://eo-virtual-archive4.esa.int.

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

## Acknowledgements

This work has been supported by the UK Natural Environment Research Council (NERC) project grant number: NE/I028017/1, which supported the lead author's Ph.D. studentship as part of the FaultLab project at the University of Leeds. The Envisat satellite data are freely available and were obtained from the European Space Agency's Geohazard Supersites project. The GNSS data were obtained from the Global Strain Rate Model project website (http://gsrm.unavco.org). Many of the figures in this paper were made using the public domain Generic Mapping Tools (GMT) software. Part of this work was carried out at the Jet PropulsionLaboratory, California Institute of Technology, under a contract with the National Aeronautics and Space Administration. COMET is the NERC Centre for the Observation and Modelling of Earthquakes, Volcanoes and Tectonics.

## Author contributions

E.H. and T.J.W. designed and wrote the paper. E.H., R.J.W., and R.L. processed the InSAR data, and D.P.S.B. and R.J.W. corrected the data for atmospheric errors. A.H. implemented the iterative unwrapping procedure into StaMPS. E.H. performed the modelling and strain rate analysis. All authors contributed to finalising and editing the paper.

## Additional information

**Competing interests:** The authors declare no competing interests.

