## [Peer Review File · Nature Communications]

Reviewers' comments:

Reviewer #1 (Remarks to the Author):

This manuscript uses space geodetic observations of interseismic deformation due to the North Anatolian Fault (NAF) in Turkey to argue that the deformation rates are relatively constant throughout much of the earthquake cycle (a few hundreds of years), except during the first ~ 10 years after large earthquakes. While the space geodetic data span a relatively short time period (10-20 years), the authors take advantage of the fact that in some places (e.g., Izmit) geodetic observations are available both before and after a large event. The fact that velocities/deformation rates return to pre-seismic values within several years after the event suggests that interseismic deformation should be a good measure of the long-term fault slip rates. The authors used simple viscoelastic earthquake cycle models to show that the quasi-constant deformation rates require relatively high effective viscosities in the lower crust, and a weaker shear zone to produce postseismic transients. The manuscript summarizes a large body of work, including analysis of the entire catalog of Envisat data spanning the NAF, and should be of interest to a broad audience. I recommend that it is published pending a moderate revision to address points below.

1) As observations of pre- and post-seismic deformation are available only on a small section of the NAF, "a complete" strain rate history for the NAF is a bit of an overstatement. I suggest modifying the title and the main text to reflect the actual observations, not deductions.

2) lines 29-35. There are alternative models of the earthquake cycle - e.g., incorporating the rate and state friction, that are able to explain available geodetic observations. Also, models assuming a linear Maxwell rheology have been shown to be inadequate in a number of studies, in particular with regard to the early vs late postseismic deformation. This would be a good place to acknowledge these issues.

3) line 87. Another relevant study of creep at Ismetpasa is: Kaneko et al., Interseismic deformation and creep along the central section of the North Anatolian fault (Turkey): InSAR observations and implications for rate-and-state friction properties, *J. Geophys. Res.*, 118, 689-697, 2013.

4) lines 98-99. Do the authors use a dislocation to model shallow creep? The assumption of a constant slip rate from the surface to the top of the seismogenic layer may be too crude, given high sensitivity of geodetic data to shallow deformation sources. It may be better to assume a stress-free boundary condition (this will also affect estimates of the shallow locking depth).

5) Figure 3. I wonder what the inferred fault slip rates and locking depths would be if only GNSS data were used in the inversions. The authors compare InSAR inversions to "model-free" GNSS estimates of fault slip rates, but presumably the GNSS data can be used in inversions as well. This bears on the issue of what new constraints do InSAR data provide on the solution compared to the GNSS data.

Reviewer #2 (Remarks to the Author):

This paper reports on a large InSAR data processing effort covering the entire North Anatolian fault in Turkey with the aim of quantifying interseismic deformation rates and patterns near the fault. The results show that short-term geodetic observations of interseismic deformation are representative of long-term interseismic deformation, unless less than a decade has passed since the last major earthquake. This result is an important result for earthquake hazard assessment, as it means we can use geodetic observations spanning a relatively small part of the earthquake cycle to assess the hazard. I have a few comments on the manuscript, most of which are minor.

I find the title a bit misleading and too bold. That interseismic deformation rates are derived for a few different times during the earthquake cycle does not mean the "complete strain rate history" is recovered. I therefore suggest to modify the title such that it describes better what is reported on in the manuscript. Note that this strong wording is repeated, e.g. in lines 38 and 48, while in line 124 it sounds more reasonable.

In figure 1 the LOS and east-west velocities are shown w.r.t. Eurasia. However, along the north coast of Turkey there are a lot of red colors in a-c indicating velocities significantly different from zeros. How is the reference picked? A single pixel location, or some average?

In line 63 it is indicated that velocities in overlapping regions are compared after projection into horizontal velocities using the average incidence angle of 23 degrees. This is somewhat surprising, as the near-range angle is about 19 degrees and far range about 26 degrees, indicating a significant (~30%) difference between how the same horizontal velocities map into near- and far-range LOS velocities. Please offer a justification for this choice (e.g. in the methods, if not here).

Some discussion about that we might still over- or under-estimate the earthquake hazard based on decade-long geodetic observations of interseismic deformation, unless we include somehow fast postseismic transient effects, would be useful.

I am not surprised the visco-elastic coupling model does so poorly. First, using Maxwell viscoelastic material has rarely been useful when researchers have had good post-seismic observations. Second, the earth structure under the upper crust is probably far from homogenous, and third, parts of the lower crust are likely elastic, but may not be brittle (i.e. elastic-ductile with some yield strength). I therefore fully agree that seismogenic depths do not well represent elastic thickness and thus also doubt that the viscosity derived here ($>10^{20}$ Pas) is of much use. Because the elastic thickness is probably much larger than the seismogenic depth and the derived locking depth, I wonder if these viscosity results hold for a model consisting of a thicker elastic lid (e.g. 30-40 km). I think a more realistic subsurface model would consist of a thicker elastic layer, containing a shear-zone under the locking depth (controlling the fast postseismic deformation) and overlying a viscous mantle layer (or halfspace).

The paragraph in lines 36-42 and the paragraph in lines 44-51 contain some overlapping information. I suggest that parts of the latter paragraph are added into the former paragraph and that the section starting in line 43 simply starts with the paragraph now starting in line 52.

Figure 1

Add information at the beginning of the caption about where in the world this is. What is the difference between tracks labeled red and black in a) and blue and black in b)? I would remove fault lines in b), as things get too busy with the GPS vectors on top. I would also label major faults, e.g. NAF and EAF and maybe distinguish somehow between primary and secondary faults, because a lot of faults are shown. Please add information about where Isparta and Izmit are, as these locations are mentioned in the text. In c), indicate what positive values mean, as "east-west velocity" does not describe this well. Move the label for Istanbul to the NW of the city to avoid covering data points.

Figure 2

Maybe mention in the first sentence that this is for the North Anatolian Fault.

Figure 3

Please spell out NAF when mentioned in this caption for the first time, as some readers only read the figure captions.

Reviewer #3 (Remarks to the Author):

In this paper, Hussain et al propose to derive an interseismic strain field along the North Anatolian Fault. This goal is motivated by the fact that different segments of this major fault lie at different stages of a roughly 250 year long earthquake cycle, allowing to explore the temporal variation of strain rates across an entire cycle (assuming a constant rheology, to first order). The main result is that strain rates are, to first order, constant over the 250 year earthquake cycle, except for a 10 year period following a large earthquake. The resulting conclusions are 1. geodetic observation can provide meaningful information when deriving seismic hazard assessments and 2. a weak fault is required to explain such behavior.

In overall, the paper is well written and reading is quite pleasant. Figures support nicely the points conveyed in the paper and I have not found major technical issues. I believe this paper is made of 2 parts. The first part is very impressive: the authors show the potential of InSAR for global strain mapping and the result is a constant strain rate along the NAF which requires a strong lower crust. This first part justifies to me the publication of this paper in a high impact journal. The second part is messy, not discussed enough and does not bring new information: we know there is a transient following earthquakes and we know several mechanisms are involved, all of them with a $\log(t)$ displacement, hence a $1/t$ decay in rates. This is a problem, as I believe studies with more advanced models already propose such conclusions. In conclusion, I think this paper proposes a very interesting approach (sampling the earthquake cycle in time with a spatial sampling) which is quite new in itself, shows a very interesting result (strain rates are constant for 240 years) but then mixes things up with speculations that are not particularly new (I detail this below). To me, either this last part should be significantly beefed up with a lot more discussion and details, either simply removed.

Detailed comments:

1. About surface strain rates. You derive a map of surface strain using the last developments of InSAR processing. The map in itself is incredibly nice and shows the potential of InSAR for global strain mapping. However, while mentioning the fact that these measures should be included in PSHA methods, there is no discussion about it. Surface strain is only a proxy of stressing rates at depth, considering some elastic properties, the degree of locking, etc, etc, making the link with PSHA not direct at all. PSHA is used directly nowadays to derive hazard maps or insurance plans, so I am afraid such light statement could lead to biased understanding from non experts who might read a Nat. Comms. paper. Furthermore, PSHA experts are already convinced about this and moving in this direction (see the last models from Jackson and Stein et al). Therefore, as you seem to imply that this is a very important point of your paper (it is in the abstract), I think it requires some additional discussion.

First, you need to clarify that including strain rates in PSHA is not as simple as it sounds. For instance, the surface strain rate that you present on figure 3d comes from your estimate of total loading rate and assuming an elastic crust. However, surface strain rates along the Ismetpasa portion of the NAF are not at all representative of what is happening at 15 km depth, but entirely controlled by shallow creep (i.e. strain is infinite at the surface there). But nonetheless, stress builds up there and this should be accounted for. So any PSHA model would be difficult to make up. Another point is that for a similar surface strain rate at the fault trace, one needs to fix either slip rate or locking depth to get a sense of stress build up at depth (i.e. the tradeoff is direct). Second (and only if you think this could be useful), you need to provide clues about how to include these measurements. If you feel that this is only a second-order conclusion of the paper, I would simply remove it.

Just a small point: it seems that geodetic strain rates agree well with seismicity over California (see line 27), but not when looking at individual regions such as Parkfield (I know the paper is not

published yet but it should be soon in BSSA, check Michel, Avouac et al) or the Katmandu area (check Ader et al 2012 and/or Stevens et al 2016). Geodesy predicts in these places that larger earthquakes are required to close the strain budget and this is not yet accounted for in PSHA.

2. About the geodetic strain rates and the long term strain rate. I agree that if the strain rates measured are representative of the long term strain rates, this rates should be used for seismic hazard assessment in some way. However, your result also shows that this 10 year period should not be taken lightly in the estimation of hazard. When integrating the strain rates over 250 years, given the log-scale used here, this plot shows that the final strain is highly influenced by the previous earthquake. What matters is the total strain increase and its spatial distribution. By integrating these strain rates over time, one should find that total strain depends in a large fraction on the post-seismic strain. So why stating that geodetic strain is an "unbiased measure of future seismic hazard"? And this is even without taking into account potential local complexities (creep, geometry, etc). Furthermore, as stated before, for a given strain rate, multiple models may be involved with very different conclusions in terms of stress build up. So there needs to be a discussion about the justification to use strain for hazard.

3. About the consistency between the data sets. When presenting the strain rates as a function of time, there is a clear inconsistency between the data points. The strain rates estimated from InSAR are derived from the surface strain predicted by the screw dislocation model. The strain rates estimated from GPS data, following the Izmit earthquake are, I suspect, derived from the difference in displacements between the two stations mentioned in the text (but this not really clear in the text). Therefore, as these stations are probably located some distance (2, 5, 10 km?) away from each other, the strain rates inferred correspond to a certain depth averaged strain rate. Could you describe this more in details? Then, it seems to me that the exponential decay is entirely driven by the post-seismic signal of Izmit, so it is difficult to accept that this is a general feature of all earthquakes along this fault. To me, post-seismic displacements with a $\log(t)$ decay are quite a general feature, but I would not bet too much about a characteristic time decay that would be constant along the fault and for all earthquakes. Please discuss this.

4. How can one compare directly and derive some grand scheme about the rheology of the crust and upper mantle when multiple mechanisms are influencing strain rates there and while only looking at the strain rates at the fault? I guess your discussion is very fuzzy here, mixing points that are well known and with others that are not clear at all.

For instance, it is clear, as shown in a paper you have written yourself, that aseismic slip affects the NAF in the vicinity of Izmit following the 1999 earthquake. Therefore, there should be, in the data, an $\log(t)$ decay function of the constitutive properties and the state of stress of the fault (if it follows a classic after slip behavior). There is also the viscous flux you mention. You do not seem to account for a potential Burger type viscosity which are apparently used a lot nowadays (even though I do not really understand why a mantle flux should have 2 characteristic times rather than 3, 4 or 15 for that matter). So there should be, at least, 2 characteristic time constants embedded in this data set. Regarding this, it is then very awkward and probably not very useful to invoke only a Maxwell viscosity and discard it based on these data. Considering the complexity and the number of mechanisms involved, there cannot be a single characteristic time and this has been know for years given all the post-seismic studies that have been published.

The conclusion about the long-term viscosity of the system being high stands, assuming a constant rheology along strike. To me, this is the most important result of the paper. Then, the fact that one needs a low viscosity is just one option among others. Aseismic slip in a perfectly elastic crust would result in the same behavior. A Burger model with a low transient viscosity of the lower crust would probably result in the same behavior.

Here, you are mixing things that we know already (there is multiple time constants because there is a lot of physical mechanisms at stake, all of them involving a $\log(t)$ behavior), things that could potentially be new (the observed behavior appears unique to the entire NAF) and grand schemes about the strength of the lithosphere that are not supported by your data. I then suggest re-writing entirely the discussion to include all potential mechanisms, what we know about post-

seismic behavior and draw some testable hypothesis for further studies (or simply remove it).

5. Given the fact that you are can derive interseismic coupling at depth, why not trying to derive a full, fault long, coupling map of the NAF? Given your results, you could derive visco-elastic Green's functions (using a Maxwell linear body with a high viscosity consistent with the long term behavior that you have shown) and use these in a simple linear inversion. It would be the first time a coupling model is derived for an entire fault in a continental domain and would be very useful for further hazard assessment. It would be a nice interpretation of the spatial variability of the very nice velocity field you have derived, using much more information than only surface strain.

6. The statement about the Omori like decay is a bit awkward as is. You should simply mention that most of the deformation mechanisms involved here have a $\log(t)$ -based decay, hence the possibility to fit these data with a $1/t$ function. Mentioning Omori sounds like you are trying to oversell this with an analogy to a famous empirical law.

To conclude, I guess it should appear more clearly that the main result is "strain rates are constant along the fault, hence during the entire earthquake cycle, apart from a 10 year transient period, which implies a long term viscosity above $1e20$ " and that the rest of the paper that follows is speculation. Once you will have provided a careful discussion about the validity of combining inconsistent pieces of data to measure surface strain, once you will have provided a careful discussion to justify why surface strain, right at the fault location, is a meaningful, useful indicator, then you will be able to conclude that some general behavior emerges from this particular fault and you will be able to speculate on the rheology of faults and the lithosphere.

A Minor Point:

Please describe in the supplementary materials the equations you are solving to derive the Maxwell strain rates on figure 4. This is less common than the arctangents you describe in great details.

I hope my comments will help improving this manuscript. I really like the approach, I find the map impressive but I am quite skeptical about the speculative part as these are not discussed enough with respect to the data available.

Reviewer #1 (Remarks to the Author):

This manuscript uses space geodetic observations of interseismic deformation due to the North Anatolian Fault (NAF) in Turkey to argue that the deformation rates are relatively constant throughout much of the earthquake cycle (a few hundreds of years), except during the first ~10 years after large earthquakes. While the space geodetic data span a relatively short time period (10-20 years), the authors take advantage of the fact that in some places (e.g., Izmit) geodetic observations are available both before and after a large event. The fact that velocities/deformation rates return to pre-seismic values within several years after the event suggests that interseismic deformation should be a good measure of the long-term fault slip rates. The authors used simple viscoelastic earthquake cycle models to show that the quasi-constant deformation rates require relatively high effective viscosities in the lower crust, and a weaker shear zone to produce postseismic transients. The manuscript summarizes a large body of work, including analysis of the entire catalog of Envisat data spanning the NAF, and should be of interest to a broad audience. I recommend that it is published pending a moderate revision to address points below.

1) As observations of pre- and post-seismic deformation are available only on a small section of the NAF, "a complete" strain rate history for the NAF is a bit of an

overstatement. I suggest modifying the title and the main text to reflect the actual observations, not deductions.

- We have amended the title and text as suggested.

Lines: 1, 45, 53, 187

2) lines 29-35. There are alternative models of the earthquake cycle - e.g., incorporating the rate and state friction, that are able to explain available geodetic observations. Also, models assuming a linear Maxwell rheology have been shown to be inadequate in a number of studies, in particular with regard to the early vs late postseismic deformation. This would be a good place to acknowledge these issues.

- We have added additional text introducing these models, as suggested.

Lines: 37-42

3) line 87. Another relevant study of creep at Ismetpasa is: Kaneko et al., Interseismic deformation and creep along the central section of the North Anatolian fault (Turkey): InSAR observations and implications for rate-and-state friction properties, J. Geophys. Res., 118, 689-697, 2013.

- We have added this reference as suggested.

Line: 101

4) lines 98-99. Do the authors use a dislocation to model shallow creep? The assumption of a constant slip rate from the surface to the top of the seismogenic layer may be too crude, given high sensitivity of geodetic data to shallow deformation sources. It may be better to assume a stress-free boundary condition (this will also affect estimates of the shallow locking depth).

- We have added text giving additional justification for our use of a simple dislocation model to account for the shallow creep. We believe this is sufficient, as the details of the creep are not the focus of this study.

Lines: 114-120

5) Figure 3. I wonder what the inferred fault slip rates and locking depths would be if only GNSS data were used in the inversions. The authors compare InSAR inversions to "model-free" GNSS estimates of fault slip rates, but presumably the GNSS data can be used in inversions as well. This bears on the issue of what new constraints do InSAR data provide on the solution compared to the GNSS data.

- In determining the slip rate and locking depths from our profiles, we did jointly invert the InSAR and the GNSS and have now clarified this in the manuscript. Both are needed particularly in areas like the central region of the NAF, where there are few to no GNSS measurements. The high spatial resolution of the InSAR also allows us to remove the impacts of shallow creep, which contaminates some of the GNSS velocities.

Lines: 107

Reviewer #2 (Remarks to the Author):

This paper reports on a large InSAR data processing effort covering the entire North Anatolian fault in Turkey with the aim of quantifying interseismic deformation rates and patterns near the fault. The results show that short-term geodetic observations of interseismic deformation are representative of long-term interseismic deformation, unless less than a decade has passed since the last major earthquake. This result is an important result for earthquake hazard assessment, as it means we can use geodetic observations spanning a relatively small part of the earthquake cycle to assess the hazard. I have a few comments on the manuscript, most of which are minor.

I find the title a bit misleading and too bold. That interseismic deformation rates are derived for a few different times during the earthquake cycle does not mean the “complete strain rate history” is recovered. I therefore suggest to modify the title such that it describes better what is reported on in the manuscript. Note that this strong wording is repeated, e.g. in lines 38 and 48, while in line 124 it sounds more reasonable.

- We have amended the title and text to reflect this concern.

Lines: 1, 45, 53, 187

In figure 1 the LOS and east-west velocities are shown w.r.t. Eurasia. However, along the north coast of Turkey there are a lot of red colors in a-c indicating velocities significantly different from zeros. How is the reference picked? A single pixel location, or some average?

- We transformed our InSAR LOS velocities for each track from a local reference area (an average of pixels in a 2~km radius) into a Eurasia-fixed GNSS reference before deconvolving into the east-west and vertical components. Any deviation from “zero” is therefore the result of atmospheric or other noise in the InSAR data.

Lines: 81-82, Supplementary Fig. S8

In line 63 it is indicated that velocities in overlapping regions are compared after projection into horizontal velocities using the average incidence angle of 23 degrees. This is somewhat surprising, as the near-range angle is about 19 degrees and far range about 26 degrees, indicating a significant (~30%) difference between how the same horizontal velocities map into near- and far-range LOS velocities. Please offer a justification for this choice (e.g. in the methods, if not here).

- The reviewer is correct and we have now recalculated the overlap regions, using the local incidence angles. This makes only a small difference to the stated uncertainties

Lines: 75, 77, 78 and Supplementary Figure S3

Some discussion about that we might still over- or under-estimate the earthquake hazard based on decade-long geodetic observations of interseismic deformation, unless we include somehow fast postseismic transient effects, would be useful.

- We have added new text in the discussion on the impacts of total strain, hence seismic hazard, based on decadal strain measurements without accounting for the postseismic transient.

Lines: 153-168

I am not surprised the visco-elastic coupling model does so poorly. First, using Maxwell viscoelastic material has rarely been useful when researchers have had good post-seismic observations. Second, the earth structure under the upper crust is probably far from homogenous, and third, parts of the lower crust are likely elastic, but may not be brittle (i.e. elastic-ductile with some yield strength). I therefore fully agree that seismogenic depths do not well represent elastic thickness and thus also doubt that the viscosity derived here ($>10^{20}$ Pas) is of much use. Because the elastic thickness is probably much larger than the seismogenic depth and the derived locking depth, I wonder if these viscosity results hold for a model consisting of a thicker elastic lid (e.g. 30-40 km). I think a more realistic subsurface model would consist of a thicker elastic layer, containing a shear-zone under the locking depth (controlling the fast postseismic deformation) and overlying a viscous mantle layer (or halfspace).

- We agree with the reviewer. The fact that we require high viscosities in the simple 2-layer model supports a scenario in which the lower crust is strong with deformation

focused in a shear zone. We have made a significant rewrite to the discussion section to reflect this and the comments of reviewer 3.

Lines: 191-193, 210-214, 221-253

The paragraph in lines 36-42 and the paragraph in lines 44-51 contain some overlapping information. I suggest that parts of the latter paragraph are added into the former paragraph and that the section starting in line 43 simply starts with the paragraph now starting in line 52.

- We have rearranged these paragraphs as suggested

Lines: 50-62

Figure 1

Add information at the beginning of the caption about where in the world this is. What is the difference between tracks labeled red and black in a) and blue and black in b)? I would remove fault lines in b), as things get too busy with the GPS vectors on top. I would also label major faults, e.g. NAF and EAF and maybe distinguish somehow between primary and secondary faults, because a lot of faults are shown. Please add information about where Ismitpeza and Izmit are, as these locations are mentioned in the text. In c), indicate what positive values mean, as “east-west velocity” does not describe this well. Move the label for Istanbul to the NW of the city to avoid covering data points.

- All suggestions by the reviewer have been implemented into Figure 1

Figure 2

Maybe mention in the first sentence that this is for the North Anatolian Fault.

- We have implemented this suggestion.

Figure 3

Please spell out NAF when mentioned in this caption for the first time, as some readers only read the figure captions.

- Done

Reviewer #3 (Remarks to the Author):

In this paper, Hussain et al propose to derive an interseismic strain field along the North Anatolian Fault. This goal is motivated by the fact that different segments of this major fault lie at different stages of a roughly 250 year long earthquake cycle, allowing to explore the temporal variation of strain rates across an entire cycle (assuming a constant rheology, to first order). The main result is that strain rates are, to first order, constant over the 250 year earthquake cycle, except for a 10 year period following a large earthquake. The resulting conclusions are 1. geodetic observation can provide meaningful information when deriving seismic hazard assessments and 2. a weak fault is required to explain such behavior.

In overall, the paper is well written and reading is quite pleasant. Figures support nicely the points conveyed in the paper and I have not found major technical issues. I believe this paper is made of 2 parts. The first part is very impressive: the authors show the potential of InSAR for global strain mapping and the result is a constant strain rate along the NAF which requires a strong lower crust. This first part justifies to me the publication of this paper in a high impact journal. The second part is messy, not discussed enough and does not bring new information: we know there is a transient following earthquakes and we know several mechanisms are involved, all of them with a $\log(t)$ displacement, hence a $1/t$ decay in rates. This is a problem, as I believe studies with more advanced models already propose such conclusions. In conclusion, I think this paper proposes a very interesting approach (sampling the earthquake cycle in time with a spatial sampling) which is quite new in itself, shows a very interesting result (strain rates are constant for 240 years) but then mixes things up with speculations that are not particularly new (I detail this below). To me, either this last part should be significantly beefed up with a lot more discussion and details, either simply removed.

Detailed comments:

1. About surface strain rates. You derive a map of surface strain using the last developments of InSAR processing. The map in itself is incredibly nice and shows the potential of InSAR for global strain mapping. However, while mentioning the fact that these measures should be included in PSHA methods, there is no discussion about it. Surface strain is only a proxy of stressing rates at depth, considering some elastic properties, the degree of locking, etc, etc, making the link with PSHA not direct at all.

PSHA is used directly nowadays to derive hazard maps or insurance plans, so I am afraid such light statement could lead to biased understanding from non experts who might read a Nat. Comms. paper. Furthermore, PSHA experts are already convinced about this and moving in this direction (see the last models from Jackson and Stein et al). Therefore, as you seem to imply that this is a very important point of your paper (it is in the abstract), I think it requires some additional discussion.

First, you need to clarify that including strain rates in PSHA is not as simple as it sounds. For instance, the surface strain rate that you present on figure 3d comes from your estimate of total loading rate and assuming an elastic crust. However, surface strain rates along the Ismetpasa portion of the NAF are not at all representative of what is happening at 15 km depth, but entirely controlled by shallow creep (i.e. strain is infinite at the surface there). But nonetheless, stress builds up there and this should be accounted for. So any PSHA model would be difficult to make up. Another point is that for a similar surface strain rate at the fault trace, one needs to fix either slip rate or locking depth to get a sense of stress build up at depth (i.e. the tradeoff is direct). Second (and only if you think this could be useful), you need to provide clues about how to include these measurements. If you feel that this is only a second-order conclusion of the paper, I would simply remove it.

Just a small point: it seems that geodetic strain rates agree well with seismicity over California (see line 27), but not when looking at individual regions such as Parkfield (I know the paper is not published yet but it should be soon in BSSA, check Michel, Avouac et al) or the Katmandu area (check Ader et al 2012 and/or Stevens et al 2016). Geodesy predicts in these places that larger earthquakes are required to close the strain budget and this is not yet accounted for in PSHA.

- We thank the reviewer for these helpful comments. As suggested, we have included an additional section in the manuscript discussing the challenges with converting surface strain measurements from geodesy into a seismic hazard assessment.

Lines: 169-185

2. About the geodetic strain rates and the long term strain rate. I agree that if the strain rates measured are representative of the long term strain rates, this rates should be used for seismic hazard assessment in some way. However, your result also shows that this 10 year period should not be taken lightly in the estimation of hazard.

When integrating the strain rates over 250 years, given the log-scale used here, this plot shows that the final strain is highly influenced by the previous earthquake. What matters is the total strain increase and its spatial distribution. By integrating these strain rates over time, one should find that total strain depends in a large fraction on the post-seismic strain. So why stating that geodetic strain is an “unbiased measure of future seismic hazard”? And this is even without taking into account potential local complexities (creep, geometry, etc). Furthermore, as stated before, for a given strain rate, multiple models may be involved with very different conclusions in terms of stress build up. So there needs to be a discussion about the justification to use strain for hazard.

- We thank the reviewer for this important point. We have added new text discussing the requirement to understand the postseismic deformation in order to determine the total strain budget.

Lines: 153-168

3. About the consistency between the data sets. When presenting the strain rates as a function of time, there is a clear inconsistency between the data points. The strain rates estimated from InSAR are derived from the surface strain predicted by the screw dislocation model. The strain rates estimated from GPS data, following the Izmit earthquake are, I suspect, derived from the difference in displacements between the two stations mentioned in the text (but this not really clear in the text). Therefore, as these stations are probably located some distance (2, 5, 10 km?) away from each other, the strain rates inferred correspond to a certain depth averaged strain rate. Could you describe this more in details? Then, it seems to me that the exponential decay is entirely driven by the post-seismic signal of Izmit, so it is difficult to accept that this is a general feature of all earthquakes along this fault. To me, post-seismic displacements with a $\log(t)$ decay are quite a general feature, but I would not bet too much about a characteristic time decay that would be constant along the fault and for all earthquakes. Please discuss this.

- We have added a section in the Methods that discusses how the GNSS strain calculations were made for the postseismic period and a justification of why we think that it represents deeper afterslip and not the shallow aseismic fault creep. We have also included a discussion of the possible impacts of the postseismic decay time constant on the overall strain and hence the hazard.

Lines: 323-328

4. How can one compare directly and derive some grand scheme about the rheology of the crust and upper mantle when multiple mechanisms are influencing strain rates there and while only looking at the strain rates at the fault? I guess your discussion is very fuzzy here, mixing points that are well known and with others that are not clear at all.

For instance, it is clear, as shown in a paper you have written yourself, that aseismic slip affects the NAF in the vicinity of Izmit following the 1999 earthquake. Therefore, there should be, in the data, an $\log(t)$ decay function of the constitutive properties and the state of stress of the fault (if it follows a classic after slip behavior). There is also the viscous flux you mention. You do not seem to account for a potential Burger type viscosity which are apparently used a lot nowadays (even though I do not really understand why a mantle flux should have 2 characteristic times rather than 3, 4 or 15 for that matter). So there should be, at least, 2 characteristic time constants embedded in this data set. Regarding this, it is then very awkward and probably not very useful to invoke only a Maxwell viscosity and discard it based on these data. Considering the complexity and the number of mechanisms involved, there cannot be a single characteristic time and this has been known for years given all the post-seismic studies that have been published. The conclusion about the long-term viscosity of the system being high stands, assuming a constant rheology along strike. To me, this is the most important result of the paper. Then, the fact that one needs a low viscosity is just one option among others. Aseismic slip in a perfectly elastic crust would result in the same behavior. A Burger model with a low transient viscosity of the lower crust would probably result in the same behavior.

Here, you are mixing things that we know already (there is multiple time constants because there is a lot of physical mechanisms at stake, all of them involving a $\log(t)$ behavior), things that could potentially be new (the observed behavior appears unique to the entire NAF) and grand schemes about the strength of the lithosphere that are not supported by your data. I then suggest re-writing entirely the discussion to include all potential mechanisms, what we know about post-seismic behavior and draw some testable hypothesis for further studies (or simply remove it).

- We have significantly rewritten and added to the discussion section of the manuscript to address the comments made by the reviewer here. We used the simple Maxwell model to understand how the system behaves. This then informs our interpretation of more complex viscous models (Burgers, power-law, depth dependent, weak zone etc).

Specifically, the inference that the rheology of the substrate (in the lower crust) away from the fault is relatively strong in the long term must hold for all viscous models.

Lines: 188, 191-193, 210-214, 221-260

5. Given the fact that you are can derive interseismic coupling at depth, why not trying to derive a full, fault long, coupling map of the NAF? Given your results, you could derive visco-elastic Green's functions (using a Maxwell linear body with a high viscosity consistent with the long term behavior that you have shown) and use these in a simple linear inversion. It would be the first time a coupling model is derived for an entire fault in a continental domain and would be very useful for further hazard assessment. It would be a nice interpretation of the spatial variability of the very nice velocity field you have derived, using much more information than only surface strain.

- We thank the reviewer for this suggestion. Although this would be interesting to pursue in further work it is beyond the scope of this paper.

6. The statement about the Omori like decay is a bit awkward as is. You should simply mention that most of the deformation mechanisms involved here have a $\log(t)$ -based decay, hence the possibility to fit these data with a $1/t$ function. Mentioning Omori sounds like you are trying to oversell this with an analogy to a famous empirical law.

- We have amended this section and removed the reference to Omori's law.

Lines: 235-236

To conclude, I guess it should appear more clearly that the main result is "strain rates are constant along the fault, hence during the entire earthquake cycle, apart from a 10 year transient period, which implies a long term viscosity above $1e20$ " and that the rest of the paper that follows is speculation. Once you will have provided a careful discussion about the validity of combining inconsistent pieces of data to measure surface strain, once you will have provided a careful discussion to justify why surface strain, right at the fault location, is a meaningful, useful indicator, then you will be able to conclude that some general behavior emerges from this particular fault and you will be able to speculate on the rheology of faults and the lithosphere.

A Minor Point:

Please describe in the supplementary materials the equations you are solving to derive the Maxwell strain rates on figure 4. This is less common than the arctangents you describe in great details.

- We have added an explicit reference to the source of the viscoelastic-coupling model equations in the Savage paper.

Line: 208-209

I hope my comments will help improving this manuscript. I really like the approach, I find the map impressive but I am quite skeptical about the speculative part as these are not discussed enough with respect to the data available.

Reviewers' comments:

Reviewer #3 (Remarks to the Author):

The authors have carefully addressed my comments and clarified what I did not previously understand. In my opinion, the paper could be accepted pending a few very minor changes.

I still have one concern and the answer could probably refine the picture you have of constant strain rates. In figure 3, you represent the along-strike distribution of slip rates and you propose a correction for the case of the Izmit post-seismic creeping region. This is perfectly fine but I believe a similar correction should be proposed for locking depth. It has been proposed that apparent locking depth varies throughout the earthquake cycle (see Jiang & Lapusta 2016). It appears that two of the locking depth estimates in the central portion of the Izmit section exceed the constant value by approximately 20-25%, while Jiang & Lapusta suggests a similar 20% change in apparent locking depth may be considered. Although deriving a precise estimate for such correction is difficult, it might be interesting to mention it and propose an alternative locking depth (and derived strain rates).

There is some re-writing required at some point. You mention that slip rates vary with increasing slip rates toward the west, meaning you believe the variation in slip rates is significant. You then mention that locking depth is constant along the 1000 km of the studied area, suggesting that variations of locking depth are not significant. Your conclusion is that strain rate is constant, despite the variation in slip rate (while strain is proportional to slip rate over locking depth). If slip rates vary, then strain rates vary. You can't have it both ways (and if you look at the strain rates east of 32°E, they decrease eastward). So I suggest a small statistical test using the results from the Monte Carlo exploration to derive the probability of having an along-strike variation of slip and strain rates.

Last line of the abstract: I would suggest replacing "of the entire lower crust" by "of the lower crust at all spatial and temporal scales", for the sake of precision.

I would move the paragraph starting on line 211 to a little later in the article. The "inescapable inference of the widespread observation of focused interseismic strain" is not only the long relaxation time away from the fault, but also the need for a rheological complexity within fault zones to explain the focusing. As you mention, your observations require the relaxation time in the lower crust far from the fault to be high. But they also require an additional rheological complexity around the fault to explain the rapid decay in strain rates near Izmit. So I would move this paragraph after you have explained the need for some fault zone complexity, even at depth. And I would remove the sentence about the spring and all, as it sounds a little like "tectonics for dummies".

Reviewer #3 (Remarks to the Author):

The authors have carefully addressed my comments and clarified what I did not previously understand. In my opinion, the paper could be accepted pending a few very minor changes.

I still have one concern and the answer could probably refine the picture you have of constant strain rates. In figure 3, you represent the along-strike distribution of slip rates and you propose a correction for the case of the Izmit post-seismic creeping region. This is perfectly fine but I believe a similar correction should be proposed for locking depth. It has been proposed that apparent locking depth varies throughout the earthquake cycle (see Jiang & Lapusta 2016). It appears that two of the locking depth estimates in the central portion of the Izmit section exceed the constant value by approximately 20-25%, while Jiang & Laputa suggests a similar 20% change in apparent locking depth may be considered. Although deriving a precise estimate for such correction is difficult, it might be interesting to mention it and propose an alternative locking depth (and derived strain rates).

While it is possible that the anomalously high locking depths on the Izmit portion of the NAF could be due to variation in locking depth through the cycle, this is complicated by the fact that the fault breaks into multiple strands here and thus widening the zone of strain accumulation. High locking depths in both pre-1999 and post-1999 profiles support the fact that the strain has split onto multiple strands.

We have now discussed this in the revised manuscript on lines: 122-126

There is some re-writing required at some point. You mention that slip rates vary with increasing slip rates toward the west, meaning you believe the variation in slip rates is significant. You then mention that locking depth is constant along the 1000 km of the studied area, suggesting that variations of locking depth are not significant. Your conclusion is that strain rate is constant, despite the variation in slip rate (while strain is

proportional to slip rate over locking depth). If slip rates vary, then strain rates vary. You can't have it both ways (and if you look at the strain rates east of 32°E, they decrease eastward). So I suggest a small statistical test using the results from the Monte Carlo exploration to derive the probability of having an along-strike variation of slip and strain rates.

We agree that increasing slip rate along strike of the NAF should imply an increasing strain rate. However, most of the slip rate increase is related to the east-west extension within central Anatolia and western Turkey. We have already corrected for this effect as described in Lines 299-313 in the Methods section. However we appreciate that this may not have been clear in the main text, so have added additional lines to clarify this point.

We have also added an additional panel in Supplementary Figure S7, showing the GNSS velocities for panel *i* (western profile) and *v* (eastern profile) on the same scale. The GNSS velocities require a slip increase from the east to the west.

Lines: 116-120, supplementary figure S7.

Last line of the abstract: I would suggest replacing “of the entire lower crust” by “of the lower crust at all spatial and temporal scales”, for the sake of precision.

Amended as suggested

Line: 19-20

I would move the paragraph starting on line 211 to a little later in the article. The “inescapable inference of the widespread observation of focused interseismic strain” is not only the long relaxation time away from the fault, but also the need for a rheological complexity within fault zones to explain the focusing. As you mention, your observations require the relaxation time in the lower crust far from the fault to be high. But they also require an additional rheological complexity around the fault to explain the rapid decay in strain rates near Izmit. So I would move this paragraph after you have explained the need for some fault zone complexity, even at depth. And I would remove the sentence about the spring and all, as it sounds a little like “tectonics for dummies”.

We have removed the clause about the spring, as suggested by the reviewer. Since the v-e coupling model can explain strain focusing without any additional rheological complexities, we have decided to leave the paragraph starting at line 221 in its current position.